# Subjective Cognitive Decline Associated with Discrimination in Medical Settings among Transgender and Nonbinary Older Adults

**DOI:** 10.3390/ijerph19159168

**Published:** 2022-07-27

**Authors:** Nickolas H. Lambrou, Carey E. Gleason, Juno Obedin-Maliver, Mitchell R. Lunn, Annesa Flentje, Micah E. Lubensky, Jason D. Flatt

**Affiliations:** 1Division of Geriatrics, Department of Medicine, University of Wisconsin School of Medicine and Public Health, Madison, WI 53792, USA; lambrou@wisc.edu (N.H.L.); ceg@medicine.wisc.edu (C.E.G.); 2Geriatric Research Education and Clinical Center, Wm S Middleton Memorial Veterans Hospital, Madison, WI 53705, USA; 3Wisconsin Alzheimer’s Disease Research Center, University of Wisconsin School of Medicine and Public Health, Madison, WI 53792, USA; 4The PRIDE Study/PRIDEnet, Stanford University School of Medicine, Palo Alto, CA 94304, USA; junoom@stanford.edu (J.O.-M.); lunn@stanford.edu (M.R.L.); 5Department of Obstetrics & Gynecology, Stanford University School of Medicine, Palo Alto, CA 94305, USA; 6Department of Epidemiology and Population Health, Stanford University School of Medicine, Palo Alto, CA 94305, USA; 7Division of Nephrology, Department of Medicine, Stanford University School of Medicine, Palo Alto, CA 94305, USA; 8Department of Community Health Systems, School of Nursing, University of California, San Francisco, CA 94143, USA; annesa.flentje@ucsf.edu (A.F.); micah.lubensky@ucsf.edu (M.E.L.); 9Alliance Health Project, Department of Psychiatry and Behavioral Sciences, School of Medicine, University of California, San Francisco, CA 94102, USA; 10Institute for Health and Aging, School of Nursing, University of California, San Francisco, CA 94143, USA; 11Department of Social and Behavioral Health, School of Public Health, University of Nevada, Las Vegas, NV 89154, USA

**Keywords:** transgender health, subjective cognitive decline, dementia risk, risk factors, social determinants of health, health disparities

## Abstract

Background: Transgender and nonbinary (TNB) individuals report greater subjective cognitive decline (SCD) compared to non-TNB people. SCD involves self-reported problems with memory and thinking and is a potential risk for Alzheimer’s disease and related dementias (ADRD). We explored psychosocial factors, such as discrimination in medical settings, associated with SCD in a sample of TNB older adults. Methods: We utilized cross-sectional data on aging health, SCD (memory complaints and worsening memory in the past year), and discrimination in medical settings from The PRIDE Study for LGBTQ+ adults aged 50+ including TNB adults (*n* = 115). Associations were tested using multivariate logistic regression. Results: Nearly 16% of TNB participants rated their memory as poor/fair, and 17% reported that their memory was worse than a year ago. TNB older adults with SCD were more likely to report experiencing discrimination in medical settings. After adjustment, those reporting discrimination in medical settings had 4.5 times higher odds of reporting worsening memory than those who did not (OR: 4.5; 95%-CI: 1.5–13.2; *p* = 0.006), and 7.5 times more likely to report poor/fair memory (OR: 7.49; 95%-CI: 1.7–32.8; *p* = 0.008); Conclusions: TNB older adults reported high frequencies of SCD and discrimination in medical settings. Further research exploring affirmative cognitive screening and healthcare services is needed.

## 1. Introduction

Research examining risk factors for Alzheimer’s disease and related dementias (ADRD) in lesbian, gay, bisexual, transgender, queer, and other-identity (LGBTQ+) populations is expanding. However, research exploring ADRD risk and resilience in transgender and nonbinary (TNB) populations is particularly sparse. The terms transgender and nonbinary represent people whose gender identity differs from commonly held notions associated with one’s sex assigned at birth. The term cisgender refers to those whose gender identity coincides with commonly held notions associated with their sex assigned at birth. Sex assigned at birth can include assessment of one’s chromosomes and/or hormone levels, but it is often determined by the outward appearance of one’s genitalia at the time of birth [1].

TNB are a particularly underserved and understudied group with distinct experiences that require prioritized attention [2]. Compared to cisgender older adults, TNB elders report higher rates of dementia diagnoses and chronic conditions that compound ADRD risk (e.g., hypertension, diabetes, obesity, stroke, mental health diagnoses) with earlier onset [3,4]. Additionally, increased levels of discrimination and marginalization experienced by TNB adults [4] are associated with poorer mental health outcomes that have damaging effects on physical health [5].

As efforts to examine ADRD in preclinical stages expand [6,7,8], so have efforts to identify markers of preclinical disease. Subjective cognitive decline (SCD) has emerged as an easily obtained and non-invasive marker that can predict progression from a cognitively healthy state to dementia, possibly serving as a marker of preclinical Alzheimer’s disease [5]. SCD refers to the self-reported perception of cognitive decline, often memory-related, and independent from objective cognitive assessment [6,9,10,11,12]. It is included in the National Institute on Aging, Alzheimer’s Association’s core clinical criteria for determining pre-dementia status [13].

Clinical samples indicate a relationship between SCD and increased risk of developing dementia [6,14]. Although rates of SCD amongst LGBTQ+ people (25%) are comparable to the general population (between 25–56%), LGBTQ+ people, as a group, are reporting SCD at younger ages (e.g., mean age 60 vs. 65) [10,15]. Our previous study found that the prevalence of SCD among TNB older adults was 17.3% (95% Confidence Interval 10.5–24.1) [16]. Considering possible endpoints associated with SCD, an analysis of 2015 Medicare claims data from the Centers for Medicare & Medicaid Services (CMS) found a higher prevalence of dementia diagnoses among transgender beneficiaries aged 65+ compared to cisgender beneficiaries (18.2% vs. 12.2%) [3].

A possible mechanism of increased ADRD risk is accelerated cognitive aging due to minority stress, which may manifest as subjective or objective cognitive decline [17]. Meyer [18] conceptualized prejudice and discrimination as minority stressors, which are: (1) unique and additive, (2) chronic, and (3) socially devised/maintained. TNB people experience discrimination at higher rates compared to cisgender people, including those who identify as cisgender LGBQ+ people [2,4]. Incurring daily discrimination and related social stressors can evoke a reaction similar to a trauma response. Repeated triggering of the autonomic nervous system (ANS) and hypothalamic–pituitary–adrenal (HPA) axis can result in accumulating and chronic ADRD risk factor burden (e.g., higher rates of diabetes, cardiovascular disease, depression) [18,19,20,21].

A complex history exists wherein TNB people have long endured discrimination and minority stressors within medical models and healthcare systems [22,23,24]. Relationships between TNB individuals and the current healthcare system are subject to negative effects of stigma, gatekeeping, provider bias, and miseducation [23]. Data reported between 2010–2015 from the National Transgender Discrimination Survey (NTDS) and US Transgender Survey revealed that one-third of TNB participants reported negative interactions with providers, including lack of provider knowledge (50%), harassment (28%), refusal of care (19%), and violence in medical settings (2%) [4,25]. Due to fear of discriminatory mistreatment, 28% of TNB participants reported avoiding healthcare altogether [4]. In another study, 40% of TNB elders reported fear of healthcare systems and providers because of discrimination and internalized stigma [26]. Furthermore, prevalence of discriminatory experiences among transgender and nonbinary people is highest amongst those who report being perceived by others as such, 40.9% and 36.9%, respectively [27].

Moreover, intersectionality of identities and social experiences (e.g., age, race/ethnicity, gender, and sexual orientation) appear to play a significant role in the likelihood of encountering discriminatory experiences in healthcare and medical settings. Data suggested that there is a complex interplay between identity, stigma, and ADRD pathology that may present barriers to early diagnosis and access to comprehensive care [11]. Higher rates of dementia for transgender TNB older adults [3] and higher rates of SCD for those who are TNB and Black, Indigenous, or People of Color compared to TNB white and cisgender Black, Indigenous, or People of Color [28,29,30] warrant further research into psychosocial factors associated with ADRD and discrimination in healthcare and medical settings.

To date, little research exists regarding relationships between psychosocial factors, SCD prevalence, and overall cognitive health for older TNB adults. This research is necessary to develop and improve dementia screening, prevention, treatment, and care for TNB populations. Using a cross-sectional national sample of TNB adults aged 50+, we examined prevalence of SCD, psychosocial factors associated with SCD (e.g., food insecurity, physical health, mental health, and discrimination in medical settings), and the relationships between discrimination in medical settings and SCD among TNB older adults. We hypothesized that SCD would be higher for TNB adults aged 50+ and that experiences of discrimination with medical settings would be associated with reporting SCD.

## 2. Materials and Methods

Rainbows of Aging (ROA) was an ancillary study of The PRIDE Study [31], a longitudinal, national, online study investigating the health of people who identify as sexual and or gender minority including but not limited to those who are lesbian, gay, bisexual, transgender, or queer (LGBTQ+). Inclusion criteria for the PRIDE Study were being 18 years of age or over, identifying as a sexual and/or gender-minority person, and residing in the US or its territories, and who were comfortable completing electronically available surveys in written English. The ROA survey focused specifically on the health of LGBTQ+ people aged 50+ (*n* = 669), and cross-sectional data were collected between August and November 2018. Inclusion criteria for the ROA module was identifying as LGBTQ+ and being aged 50+. No exclusion criteria were applied. ROA data were examined for this analysis, which focused on participants who are aged 50+, and self-identified as TNB of any sexual orientation (*n* = 115) or cisgender LGBQ+ (*n* = 592).

Subjective cognitive decline (SCD) measures were self-reported, and mirrored questions in the Health and Retirement Study [14,32,33,34]. Two items measured SCD: (1) “How would you rate your memory at the present time? Would you say it is excellent, very good, good, fair, or poor?” with responses ranging from 0 (poor) to 4 (excellent), and (2) “Compared to a year ago, would you say your memory is better now, about the same, or worse now than it was then?” with scores ranging from (1) worse, (2) same, and (3) better. Discrimination was assessed using questions from the MacArthur Foundation Midlife Development in the United States (MIDUS) survey, a population-based survey that examined experiences of discrimination in medical settings. [35] Discrimination questions in this study included reporting overall discrimination (harassment/name calling from strangers in public; yes/no), reporting physical violence from others due to gender identity (yes/no), and reporting discrimination in a medical setting (yes/no). Additional psychosocial factors measured by self-report included questions on food insecurity (e.g., reporting that you ate less than you felt you should because there wasn’t enough money to buy food in the past 12 months); ratings of physical health and emotional health (poor, fair, good, very good, or excellent) [14]; a previous diagnosis of a mental health condition (e.g., depression and posttraumatic stress disorder). Demographic data included questions about age, education (less than high school, high school/GED, or some college or higher), and racial identity (select all that apply) and included American Indian or Alaska Native, Asian, Black or African-American, Hawaiian or other Pacific Islander, White, and/or Another race. Sex assigned at birth options were female and male. For gender identity, ‘select all that apply’ options included man, nonbinary/genderqueer, transgender man, transgender woman, woman, or another gender identity.

We used descriptive statistics to characterize demographic, psychosocial factors, and SCD prevalence. Associations between discrimination in medical settings and SCD were examined using chi-square and *t*-tests. Multivariate logistic regression was used to test associations between SCD and health, psychosocial factors, and discrimination, with results presented as odds ratios (OR) and 95% confidence intervals (CIs). Only health and psychosocial factors that were significantly associated with SCD (*p*-value < 0.05) in bivariate analyses were included in the final models. We limited final regression models to testing the associations between SCD and psychosocial factors among TNB participants only (*n* = 115) given past research finding higher rates of SCD in TNB adults [15]. All models were adjusted for age (years) and education (less than or equal to high school/GED vs. some college or higher). All analyses were conducted using SPSS (Statistical Package for Social Sciences) version 22.0 (IBM, Armonk, NY, USA).

## 3. Results

### 3.1. Demographic Characterstics

TNB participants included those who reported their gender identity as transgender man, transgender woman nonbinary/genderqueer, or any/many gender identity(ies) that were different from their sex assigned at birth. For sexual orientation, all participants could select any that apply, and options included Asexual, Bisexual, Gay, Lesbian, Pansexual, Queer, Questioning, Same-gender loving, Straight/Heterosexual, and/or another sexual orientation.

Median age for TNB participants in years was 58.2, range: 50–76. Participants were well-educated with more than 95% reporting attending some college or higher, and predominantly white (90%). In terms of TNB gender identity, 45% of participants identified as transgender men, 35% genderqueer, 20% transgender women, and 12% another gender identity (Table 1). Cisgender LGBQ+ older adults did not differ significantly from TNB in key demographic characteristics.

### 3.2. Subjective Cognitive Decline

Next, we explored differences in SCD by TNB and cisgender LGBTQ+ in terms of poor/fair memory vs. good/very good/excellent and worsening memory vs. no change. Nearly 16% of TNB participants rated their memory as poor/fair compared to 12% of LGBQ+ participants (Table 1).

Next, we explored demographic and psychosocial factors associated with the SCD outcomes in TNB older adults only by comparing those with and without memory problems (Table 2). TNB participants rating their memory as poor/fair were more likely to report food insecurity (44% vs. 19%, *p* = 0.03), poor/fair physical health status (50% vs. 19%, *p* = 0.01), discrimination in medical settings related to gender identity (82% vs. 32%, *p* < 0.001), and physical violence from others related to gender identity (71% vs. 38%, *p* < 0.01) than those who rated their memory as good/very good/excellent. There were no differences between poor/fair memory and poor/fair emotional health as well as a past diagnosis of depression and PTSD. Experiencing discrimination in medical settings (Table 3) was the only factor associated with worsening memory (68% vs. 33%, *p* = 0.004).

### 3.3. Discrimination in Medical Settings and Subjective Cognitive Decline

After accounting for age, race/ethnicity, education, and significant psychosocial factors identified from bivariate analyses, TNB people who experienced discrimination in medical settings (Table 4) had 7.5 times greater odds of reporting poor/fair memory (OR: 7.49; 95%-CI: 1.7–32.9; *p* = 0.008) and 5.3 times greater odds of experiencing worsening memory over the past year (OR: 5.33; 95%-CI: 1.78–16.0; *p* = 0.003) than those who did not experience discrimination in medical settings.

## 4. Discussion

This study examined SCD and its association with psychosocial factors, such as mental health diagnoses, violence, and discrimination in medical settings in a sample of TNB older adults. Around 16% of TNB older adults reported SCD, and TNB participants reporting poor/fair memory and worsening memory were more likely to report having experienced discrimination in medical settings, even when accounting for age, education, and other psychosocial factors. Altogether, these data suggested that TNB older adults may face unique barriers that may prevent them from accessing medical care and talking with healthcare providers about their memory problems.

Our findings suggest that 16% to 17% of TNB older adults reported problems with their memory, which is similar to a previous population-based study reporting the SCD prevalence of 17.3% (95% Confidence Interval 10.5–24.1) among TNB adults aged 45+ [16]. Prevalence estimates of SCD among the general population range from 15% to 60% in adults aged 65 and older [15,36]. Important to note is that the median age of TNB participants in this study was 58.2 years. This median age, lower than many studies of older adults, could suggest that TNB elders may report SCD at younger ages and could benefit from educational programs on brain and cognitive health and cognitive screenings [15], especially in light of literature reflecting higher rates of dementia amongst transgender adults compared to cisgender adults [3]. Future research on dementia, SCD, access to medical care, and related psychosocial factors among TNB older adults is needed.

Interestingly, while TNB older adults reported high rates of a past depression diagnosis, this was not associated with reporting memory problems. This finding is contrary to other studies that found an association between depressive symptoms and SCD [9,36,37,38]. However, this may be due to the smaller sample size of TNB with SCD in our study. Our findings indicated that SCD was associated with several psychosocial stressors, such as economic strain, food insecurity, and experiencing violence.

Despite diagnostic changes over time reflecting healthcare’s shifting views of TNB identities [39,40], discrimination in medical settings was significantly associated with SCD for TNB older adults in our study. Overall, TNB adults were over two times more likely to report discrimination in medical settings compared to their cisgender LGBQ+ peers in our study. TNB older adults who reported experiencing significant discrimination in healthcare and medical settings were 5 to 8 times more likely to report worsening memory and poor/fair memory. These findings support the need for research focused specifically on ways to promote cognitive health screenings and discussions with healthcare professionals among TNB older adults experiencing SCD. Additionally, strategies to eliminate discrimination in medical settings for TNB older adults could be important for promoting cognitive health, increasing early detection of dementia, and reducing related health disparities [23].

The historical context of discrimination experienced by TNB people in healthcare and medical settings underpins our current difficulties in providing affirmative care [23]. Terms such as transgender and cisgender are derived from assumptions based on one’s sex assigned at birth. These assumptions are derived from essentialist frameworks, such that identity is fixed, one will internally identify their gender (or gender identity), outwardly express their gender identity (gender expression), and be attracted to others (sexual orientation) in heteronormative and binary fashion in accord with social norms.

In other words, TNB individuals have the added burden of trying to receive competent care in a system that has historically excluded them, and educating providers around TNB identity and health has been indicated as a significant and persistent stressor [4,23]. A qualitative study on healthcare experiences among transmasculine identifying people revealed specific recommendations for systemic change in research and clinical settings that can help reduce this burden, including: (1) amending binary items and language on medical forms, (2) making spaces more accessible (e.g., all-gender restrooms), and (3) encouraging providers to educate themselves and explore personal and professional biases [23]. There is a need to improve access to healthcare and ensure that TNB adults receive inclusive and welcoming care when accessing aging-related support services.

Disparate rates of SCD and dementia combined with discrimination in healthcare and medical settings for TNB people with SCD is concerning, given that our past research has found that more than 50% of LGBTQ+ people with SCD do not talk to their healthcare providers about their SCD [16]. Overall, more than half of older adults with dementia do not receive a formal diagnosis [41]. However, discrimination in healthcare and medical settings compounds existing barriers. There are several benefits of early dementia diagnosis. This includes early initiation of care plans and treatments, prevention of unsafe behaviors (e.g., driving, unsafe cooking practices, medication non-adherence), and subsequent hospitalizations. Moreover, early diagnosis can help rule out modifiable contributors to SCD, such as medication side effects, infections, sleep problems, and treatment of depression [42,43,44]. Given our findings, additional training of healthcare providers and facilitating systemic and policy change in medical settings are needed to ensure that TNB adults, with and without SCD, can receive inclusive and competent care as they age.

## 5. Limitations

This was an exploratory study, and more research is needed to clarify relationships between psychosocial factors, discrimination, and SCD for TNB people. While minority stress and lack of provider education are two factors underpinning outcomes, contributors to this study’s findings are likely multifaceted. Secondly, this was a relatively small sample of TNB people aged 50+ (*n* = 115), with most participants being White and well-educated, potentially limiting relevancy to additional populations. Another limitation of our study is that we relied on self-report. In terms of cognitive decline, studies have found self-report to be valid measure for reporting other conditions (e.g., diabetes, hypertension, stroke) [45] and that individuals reporting SCD are two times more likely to develop dementia compared to those without SCD [46]. In addition, varying levels of self-disclosure and the intersection of any number of identities make for a complex picture, substantiating the need for systemic change and future research.

## 6. Future Directions

Research examining experiences related to intersectionality, SCD, and discrimination in healthcare and medical settings, with larger samples of TNB people of color, immigrants, and Indigenous communities is greatly needed. For example, future research would benefit from models exploring how intersecting marginalized identities relate to the prevalence of SCD and/or discriminatory experiences in medical settings. Increased representation and diversity in aging studies that specifically examine SCD and dementia risk in TNB populations is also necessary, enabling future research to reflect outcomes within TNB subgroups. Moreover, given the sociopolitical context of this generational cohort, exploration of physical and psychological factors related to the HIV epidemic may be particularly salient. Another useful direction includes exploration of internal and external expectations regarding mental abilities amongst TNB and other marginalized populations compared to those non-marginalized identities.

## 7. Conclusions

This research serves as a foundation for a deeper understanding of SCD and associated psychosocial factors in TNB populations. Psychosocial factors, such as poor health, food insecurity, violence, and discrimination in medical settings were associated with SCD. Further research exploring SCD and the complex relationships between modifiable psychosocial factors for cognitive health in TNB older adults would be beneficial to reducing health disparities amongst this population.

In addition, given that TNB elders are reporting higher rates of SCD compared to the general population [16], it is highly concerning this group also reports high rates of discrimination in healthcare and medical settings. Discrimination in healthcare and medical settings will serve as an additional barrier in obtaining dementia diagnostics, care, and support services. Discriminatory practices in healthcare and medical settings could be viewed as modifiable factors for healthcare providers and targeting such practices could lead to better care and dementia diagnostics amongst diverse populations [23,26,47]. This requires a shift in organizational culture, including prioritizing competency training in TNB healthcare, modernizing research methods to include TNB participants, and recruiting and retaining a diverse and competent workforce in healthcare research and practice [48,49].

## Figures and Tables

**Table 1 ijerph-19-09168-t001:** Demographic Characteristics for Transgender and Nonbinary and Cisgender Lesbian, Gay, Bisexual, Queer, and Another Identity Older Adults (*n* = 115).

Variable	TNB (*n* = 115) n (%)	Cisgender LGBQ+ (*n* = 592) n (%)
**Age, Median; Range**	58.2; Range: 50–76	59.7; Range: 50–82
**Gender identity**		
Another gender identity	14 (12.2)	-
Genderqueer/Nonbinary	40 (34.8)	-
Man	22 (19.1)	378 (63.5)
Transgender man	23 (45.2)	-
Transgender woman	52 (20.0)	-
Woman	-	216 (36.5)
**Sex assigned at birth**		
Female	46 (40.0)	216 (36.5)
Male	68 (59.6)	378 (63.5)
**Sexual orientation**		
Another sexual orientation	27 (23.5)	11 (1.9)
Asexual	25 (21.7)	10 (1.7)
Bisexual	34 (29.6)	56 (9.5)
Gay	18 (15.7)	374 (63.2)
Lesbian	35 (30.4)	154 (26.0)
Pansexual	19 (16.5)	15 (2.5)
Straight/Heterosexual	10 (8.7)	7 (1.2)
Queer	33 (28.7)	42 (7.1)
**Race/Ethnicity**		
American Indian or Alaska Native	7 (6.1)	15 (2.5)
Another race/ethnicity	4 (3.5)	12 (2.0)
Asian	2 (1.7)	9 (1.5)
Black or African-American	4 (3.5)	12 (2.0)
Latino	8 (7.3)	37 (6.3)
Person of color	19 (16.5)	75 (12.7)
White	103 (89.6)	527 (89.0)
**Education**		
Some college or higher	111 (95.5)	573 (96.8)
**Subjective Cogntive Decline**		
Memory—Poor/Fair	18 (15.8)	68 (11.9)
Worsening memory	19 (16.7)	87 (15.2)

*Note.* Sexual orientation, gender identity, and race/ethnicity were ‘select all that apply’ and may not add up to 100%. Person of color was anyone who selected an exclusively non-white identity (Latino, American Indian or Alaska Native, Black or African American, Another race/ethnicity, or Asian).

**Table 2 ijerph-19-09168-t002:** Psychosocial Factors Associated with Current Self-reported Poor/Fair Memory in Transgender and Nonbinary Older Adults.

Characteristic	No Memory Problems *n* = 96 (84.2%)	Poor to Fair Memory *n* = 18 (15.8%)	*p*-Value
Age, median	59.6	58.6	0.54
Person of color	15 (15.6)	4 (22.2)	0.50
Food insecurity	18 (18.8)	8 (44.4)	**0.03** *
Poor to fair physical health	18 (18.9)	9 (50.0)	**0.01** *
Poor to fair emotional health	29 (30.2)	7 (38.9)	0.47
Depression	64 (66.7)	15 (83.3)	0.16
PTSD	30 (31.6)	10 (55.6)	0.05
Discrimination	69 (72.6)	16 (94.1)	0.07
Physical Violence	36 (37.9)	12 (70.6)	**0.01**
Discrimination in medical settings related to gender identity	30 (31.6)	14 (82.4)	**<0.001** *

*Note.* Person of color was anyone who selected an exclusively non-white identity. Boldface * indicates statistical significance. PTSD = post traumatic stress disorder.

**Table 3 ijerph-19-09168-t003:** Psychosocial Factors Associated with Worsening Memory in the Past Year Among Transgender and Nonbinary Older Adults.

Characteristic	Same/Better *n* = 95 (83.3%)	Worse Memory *n* = 19 (16.7%)	*p*-Value
Age, median	59.3	59.7	0.81
Person of color	18 (18.9)	1 (5.3)	0.19
Food insecurity	20 (21.1)	6 (31.6)	0.37
Poor to fair physical health	20 (21.3)	7 (36.8)	0.15
Poor to fair emotional health	31 (32.6)	5 (26.3)	0.59
Depression	63 (66.3)	16 (83.3)	0.12
PTSD	32 (34.0)	8 (42.1)	0.50
Discrimination	70 (75.3)	15 (78.9)	0.73
Violence	56 (60.2)	8 (42.1)	0.15
Discrimination in medical settings due to gender identity	31 (33.3)	13 (68.4)	**0.004** *

*Note.* TNB represents transgender and nonbinary gender identities. Person of color was anyone who selected an exclusively non-white identity. Boldface * indicates statistical significance.

**Table 4 ijerph-19-09168-t004:** Logistic Regression Models Examining Association Between Discrimination in Medical Settings and Subjective Cognitive Decline in Transgender and Nonbinary Older Adults.

Characteristics	Poor to Fair Memory ^a^			Worsening Memory ^b^		
	OR	95% CI	*p*-Value	OR	95% CI	*p*-Value
Age	1.03	0.93, 1.14	0.58	1.01	0.93, 1.10	0.55
Education	0.44	0.11, 1.73	0.24	0.85	0.27, 2.71	0.68
Poor to fair health	3.53	0.91, 13.72	0.07	-	-	-
Food insecurity	1.42	0.42, 4.89	0.57	-	-	-
Violence	1.51	0.40,5.69	0.55	-	-	-
Discrimination in medical settings due to gender identity	7.49	1.71, 32.79	**0.008** *	4.48	1.53, 13.17	**0.006** *

*Note.* TNB represents transgender and nonbinary gender identities. Person of color was anyone who selected an exclusively non-white identity. Boldface * indicates statistical significance. CI = confidence interval. Model information: ^a^ χ^2^ (df = 6, *n* = 111) = 21.02, *p* = 0.002; Nagelkerke R^2^ = 0.30; ^b^ χ^2^ (df = 3, *n* = 111) = 8.40, *p* = 0.04; Nagelkerke R^2^ = 0.12.

## Data Availability

The data presented in this study are available on request from The PRIDE Study at https://pridestudy.org/collaborate (accessed on 13 May 2022). The data are not publicly available to protect confidentiality of the research participants.

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
