# Peer review of "Subjective Cognitive Decline Associated with Discrimination in Medical Settings among Transgender and Nonbinary Older Adults"

_ijerph, 2022, doi:10.3390/ijerph19159168_

Round 1

Reviewer 1 Report

Thank you for the opportunity to review ‘Subjective Cognitive Decline associated with discrimination in medical settings among transgender and nonbinary older adults’. The paper is well-researched and well-presented, and presents findings from the longitudinal PRIDE study in the US (N=669), of whom 115 identified as trans- or non-binary. The authors find that TNB participants reported subjective cognitive decline at a higher rate than do sexual minorities (LGBQ+). The authors associate the higher rate of SCD with experiences of discrimination in medical settings.

Although I commend the authors for a well-presented paper there are a couple of missing pieces for me that I would need to see to be converted to the authors’ argument. Firstly, it is not specified how many of the ‘LGBQ+’ were open and disclosing about their identities to their health care providers. There is abundant research (including my own) that suggests that LGBQ+ people do not universally disclose their sexualities to their health care providers; in fact gay/bi men in particular (and especially older ones) are far less likely to disclose than any other subgroup (lesbian/bi women are more likely to disclose because of the birth control question). I have not been in the room when a trans person presents for medical care, but I would suspect that it would be quite difficult for a trans person not to disclose to their health care providers. Thus, discrimination (and stigma—which in my opinion is far more insidious and more difficult to capture than the legal category of discrimination) would be more likely against gender diverse persons than sexually diverse persons simply because of the ability of the latter to ‘pass’, or at least not disclose (which might readily explain the finding in line 257 of the paper). If the authors were to compare disclosing TNB with disclosing LGBQ+ persons I wonder if the differences in experience of stigma and of reported SCD would sustain? The authors need to compare like with like.

Secondly (and stay with me for a moment), the data are from 2018 (line 130), which of itself is perfectly fine. People aged 50-76 in 2018 would been born between 1942-1968; those of the mean age of 58 would have been born in 1960. This puts this cohort between the sexually active years of 20 and 30 years of age during 1980-1990, the first (and pre-anti-retroviral treatment) decade of the HIV epidemic in the US. A signal presentation of HIV in that era was cognitive decline and subcortical dementias. Is it not possible that all gender and sexually diverse persons who lived through the early decades of AIDS/HIV would be hypervigilant to cognitive changes, real or perceived? Is it not possible that persons who have experienced stigma and discrimination learned to have very high expectations of their mental abilities, and would therefore be very aware of (and concerned about) even minor real or perceived changes?  Is the overall mental performance bar for them higher than for the cis-het population?

I saw no reference to HIV in the paper, and that itself is concerning, since HIV can and does contribute to mental health changes. Of those of us who lived through that era, who wasn’t depressed at some point? Have we moved so far away from that experience that we no longer consider it important? It surely is for this age group.

This study identifies the differences in these populations but takes note only of apparent associations. I completely support the recommendation to improve health care provider education. That is a familiar trope. But I wonder if provider attitude is a necessary but insufficient explanation of the differences that the authors have identified. If this article is to be a meaningful contribution to the literature it cannot merely say ‘this is one more example of minority stress’, and ‘health care providers need to do better’. These things are true, of course, but I’m not convinced—at least from the data presented in this study, and by the theoretical framing of those data—that there are not multifarious contributors to the findings. Neither the PRIDE study nor the authors may have the data or resources to hypothesise about these more complex associations, but would it not be feasible to raise these (or dismiss them) as possibilities in the discussion and limitations sections of the paper?  

The authors describe stressors as additive (line 85), and also report that Black and Native Americans experienced ‘discrimination’ at higher rates than other racial groups; it would be interesting for the authors to use their data to build a model to see how each stigmatised status (gender diversity, sexual diversity, racial identity) contributes to the overall experience of discrimination. That might be beyond the scope of the present paper, but it is the authors who raise this issue, and any person of colour will certainly have significant stories of discrimination and stigma to tell. How do these intersectional (or additive) factors intersect (or add up)?

Did I mention a limitations section? There isn’t one. This is an essential addition to this paper, and would be an opportunity to explore the limits of the conclusions and the theorising about this topic. It would also be a place for the authors to say something about the next steps for research in this area.

This is a good paper as far as it goes, and I hope the authors will take the opportunity to make it great one.

Author Response

Response to Reviewers:

We thank the reviewers for their thoughtful comments, which have helped us to strengthen the paper. Please see our point-by-point responses below. All page numbers referenced are for the tracked changes version of the revised manuscript. We have also made editorial changes throughout the paper to improve clarity.

Reviewer 1:

  1. Although I commend the authors for a well-presented paper there are a couple of missing pieces for me that I would need to see to be converted to the authors’ argument. Firstly, it is not specified how many of the ‘LGBQ+’ were open and disclosing about their identities to their health care providers. There is abundant research (including my own) that suggests that LGBQ+ people do not universally disclose their sexualities to their health care providers; in fact gay/bi men in particular (and especially older ones) are far less likely to disclose than any other subgroup (lesbian/bi women are more likely to disclose because of the birth control question). I have not been in the room when a trans person presents for medical care, but I would suspect that it would be quite difficult for a trans person not to disclose to their health care providers. Thus, discrimination (and stigma—which in my opinion is far more insidious and more difficult to capture than the legal category of discrimination) would be more likely against gender diverse persons than sexually diverse persons simply because of the ability of the latter to ‘pass’, or at least not disclose (which might readily explain the finding in line 257 of the paper). If the authors were to compare disclosing TNB with disclosing LGBQ+ persons I wonder if the differences in experience of stigma and of reported SCD would sustain? The authors need to compare like with like.

Response: We appreciate the point about the need to understand disclosing of identities. We highlight under the methods section that we included the Rainbows of Aging module, which focused on the health of LGBTQ+ people aged 50+ and 669 respondents who identified as LGBTQ+ completed the study. We do not have any information on participants in the study that did not disclose that they identified as LGBTQ+ as this was part of the parent study eligibility. We do not know about limited disclosure of TNB identities from our study. Of the 669 respondents, 115 participants identified as TNB. We have revised the methods to reflect that the modules for this study was part of a larger study, The PRIDE Study, which is investigating the health of the LGBTQ+ community. Please see revisions on page 3 lines 128-130.

  1. Secondly (and stay with me for a moment), the data are from 2018 (line 130), which of itself is perfectly fine. People aged 50-76 in 2018 would been born between 1942-1968; those of the mean age of 58 would have been born in 1960. This puts this cohort between the sexually active years of 20 and 30 years of age during 1980-1990, the first (and pre-anti-retroviral treatment) decade of the HIV epidemic in the US. A signal presentation of HIV in that era was cognitive decline and subcortical dementias. Is it not possible that all gender and sexually diverse persons who lived through the early decades of AIDS/HIV would be hypervigilant to cognitive changes, real or perceived? Is it not possible that persons who have experienced stigma and discrimination learned to have very high expectations of their mental abilities, and would therefore be very aware of (and concerned about) even minor real or perceived changes? Is the overall mental performance bar for them higher than for the cis-het population?

I saw no reference to HIV in the paper, and that itself is concerning, since HIV can and does contribute to mental health changes. Of those of us who lived through that era, who wasn’t depressed at some point? Have we moved so far away from that experience that we no longer consider it important? It surely is for this age group.

Response: We appreciate the points about the impact of the HIV epidemic in the U.S. We did not focus on HIV given that few TNB participants endorsed HIV positive status (n=5) and we did not have statistical power to explore differences among TNB older adults. We have included in “Limitations” and “Future Directions” sections on page 8 the need for future research that explores impact of the HIV epidemic physically and psychologically among older TNB populations.

  1. This study identifies the differences in these populations but takes note only of apparent associations. I completely support the recommendation to improve health care provider education. That is a familiar trope. But I wonder if provider attitude is a necessary but insufficient explanation of the differences that the authors have identified. If this article is to be a meaningful contribution to the literature it cannot merely say ‘this is one more example of minority stress’, and ‘health care providers need to do better’. These things are true, of course, but I’m not convinced—at least from the data presented in this study, and by the theoretical framing of those data—that there are not multifarious contributors to the findings. Neither the PRIDE study nor the authors may have the data or resources to hypothesise about these more complex associations, but would it not be feasible to raise these (or dismiss them) as possibilities in the discussion and limitations sections of the paper?

Response: We appreciate the points about potential impact of provider attitudes. Unfortunately, we only have data on participants’ views. In the limitations of the paper, we highlight the need to also address provider attitudes in additional to provider knowledge/education.

  1. The authors describe stressors as additive (line 85), and also report that Black and Native Americans experienced ‘discrimination’ at higher rates than other racial groups; it would be interesting for the authors to use their data to build a model to see how each stigmatised status (gender diversity, sexual diversity, racial identity) contributes to the overall experience of discrimination. That might be beyond the scope of the present paper, but it is the authors who raise this issue, and any person of colour will certainly have significant stories of discrimination and stigma to tell. How do these intersectional (or additive) factors intersect (or add up)?

Response: We agree with the recommendation to address intersectionality and how race/ethnicity may also contribute to experiences of discrimination. Given the study sample is 17% or n=19 identified as a person of color. In the limitations/future directions, we have added the need to explore intersectionality and experiences given any particular combination of marginalized identities may be important to explore.

  1. Did I mention a limitations section? There isn’t one. This is an essential addition to this paper, and would be an opportunity to explore the limits of the conclusions and the theorising about this topic. It would also be a place for the authors to say something about the next steps for research in this area.

This is a good paper as far as it goes, and I hope the authors will take the opportunity to make it great one.

Response: Your excellent feedback outlined above led to the key additions of dedicated “Limitations” and “Future Directions” sections on page 8. You brought several insightful and critical items that needed to be named as limitations and directions for future research, such as:

  • Levels of self-disclosure in medical settings and associations with discrimination
  • While minority stress and lack of provider education are two factors underpinning outcomes, contributors to this study’s findings are likely multifaceted
  • Intersectionality and experiences given any particular combination of marginalized identities
  • Impact of the HIV epidemic physically and psychologically within this generational cohort
  • Internal and external expectations of cognitive abilities of those with marginalized identities compared to those with non-marginalized identities.

Reviewer 2 Report

This manuscript describes compilation of results from the PRIDE Study, specifically related to subjective cognitive decline (SCD) in transgender and non-binary older adults (TBD).  The authors extracted information from the cross-sectional PRIDE Study and analyzed data related to SCD in the desired study population, and determined several relationships between SCD and TNB, specifically higher incidence of SCD, and a relationship between SCD and healthcare discrimination.

Points to address:

1. The introduction section is quite long, and includes many pieces of information and literature review that would be better served in the Discussion section.  Recommend the authors review and relocate the relevant sources and discussion.

2. There is no hypothesis.  The author mention their purpose was to examine various relationships for correlation, however they did not commit to a hypothesis - this is a major weakness and must be addressed.

3. In the Materials and Methods section the authors begin by describing the PRIDE Study, rather than their own methods for the current study.  Would recommend first describing their methods, then using the PRIDE Study information to further explain from where they obtained their data.  This manuscript is not about the PRIDE Study in general, so should be edited accordingly.

4. Also in M&M section - lines 169-171 describe results rather than materials and methods.  This should be moved to the Results section.

5. Results:  demographic information revealed the study population was well-educated and primarily white.  This should be addressed as a limitation and contributor toward possible selection bias, as it could be a contributor to results, as well as relevancy to additional populations.

6. Results:  lines 194-197 state that although not statistically significant, reports of SCD were higher for TNB.  Recommend removing this statement, as the statistical analysis shows this was NOT SS and saying that even so the response was subjectively higher could confuse readers.

7. Discussion:  lines 230-232 say that factors such as mental health problems etc were associated with SCD.  I would adjust the wording to reflect that factors including these diagnoses were associated, rather than giving them as examples of possible factors, as the data does support the fact that the stated factors WERE associated.

8. Discussion:  the information stated in lines 245-247 was already provided in the Intro.  Would remove from intro, as it is more appropriate for Discussion.

9. Discussion:  the paragraph spanning lines 267-273 gives historical context rather than current data adding to the discussion of the results.  As such would either remove entirely, or relocated to the Introduction.

10. Although the authors hint at some limitations these are not specifically addressed or named as such.  Recommend a specific limitations and strengths section so there is no confusion by readers.  The also allows the concerns and limitations to be addressed by the authors themselves.

Overall I believe this manuscript describes research that is relevant and important; with the edits and changes above, I recommend publications.

Author Response

Response to Reviewers:

We thank the reviewers for their thoughtful comments, which have helped us to strengthen the paper. Please see our point-by-point responses below. All page numbers referenced are for the tracked changes version of the revised manuscript. We have also made editorial changes throughout the paper to improve clarity.

  1. The introduction section is quite long, and includes many pieces of information and literature review that would be better served in the Discussion section. It is recommended the authors review and relocate the relevant sources and discussion.

            Response: We appreciate this recommendation. We felt that providing background on subjective cognitive decline, discrimination and intersectionality was necessary to guide readers and establish the importance of the proposed research. We have made some editorial changes to the paper to help to streamline the background.

  1. There is no hypothesis.  The author mention their purpose was to examine various relationships for correlation, however they did not commit to a hypothesis - this is a major weakness and must be addressed.

            Response: We agree with the review that a hypothesis was not stated. We now add our original hypothes on page 3 lines 122-123 “We hypothesized that SCD would be higher for TND adults aged 50+ and that experiences of discrimination with medical settings would be associated with reporting SCD.”

  1. In the Materials and Methods section the authors begin by describing the PRIDE Study, rather than their own methods for the current study. Would recommend first describing their methods, then using the PRIDE Study information to further explain from where they obtained their data.  This manuscript is not about the PRIDE Study in general, so should be edited accordingly.

Response:  As recommended, in the Materials and Methods section, we edited to begin by describing the ROA module, followed by the module’s inclusion in the PRIDE study. You will find these edits reflected on pages 3 & 4.

  1. Also in M&M section - lines 169-171 describe results rather than materials and methods.  This should be moved to the Results section.

Response:  As recommended, we moved these lines to the first paragraph of the Results section. This edit can be found on page 4.

  1. Results:  demographic information revealed the study population was well-educated and primarily white.  This should be addressed as a limitation and contributor toward possible selection bias, as it could be a contributor to results, as well as relevancy to additional populations.

Response:  As recommended, we expanded upon this limitation to caution against potential relevancy of results to additional populations. We also added the need for larger samples of TNB people of color in a new Future Directions section. These edits can be found on page 8.

  1. Results:  lines 194-197 state that although not statistically significant, reports of SCD were higher for TNB.  Recommend removing this statement, as the statistical analysis shows this was NOT SS and saying that even so the response was subjectively higher could confuse readers.

Response:  As recommended, this statement was removed, as to not confuse readers. The deletion of this statement is reflected on page 5.

  1. Discussion:  lines 230-232 say that factors such as mental health problems etc were associated with SCD.  I would adjust the wording to reflect that factors including these diagnoses were associated, rather than giving them as examples of possible factors, as the data does support the fact that the stated factors WERE associated.

Response:  As recommended, we adjusted the wording reflect that factors including these diagnoses were associated. This edit appears in the first paragraph of the Discussion on page 6.

  1. Discussion:  the information stated in lines 245-247 was already provided in the Intro.  Would remove from intro, as it is more appropriate for Discussion.

Response: While we did not move this information from the Discussion section, we did make edits to better situate our data in the literature rather than simply being redundant. This edit appears on page 7.

  1. Discussion:  the paragraph spanning lines 267-273 gives historical context rather than current data adding to the discussion of the results.  As such would either remove entirely, or relocated to the Introduction.

Response: It was recommended to remove lines only focusing on historical context (rather than current data) entirely, or relocate to the Introduction. We edited the narrative to be better suited for the discussion section by tying it more closely with current study implications and the wider body of existing literature. This edit is reflected on page 7.

  1. Although the authors hint at some limitations these are not specifically addressed or named as such.  Recommend a specific limitations and strengths section so there is no confusion by readers.  The also allows the concerns and limitations to be addressed by the authors themselves.

Overall I believe this manuscript describes research that is relevant and important; with the edits and changes above, I recommend publications.

Response:  As recommended, we added a specific Limitations section, along with a Future Directions section on page 8. These additions were critical, and thank you for all of your clear and valuable feedback!

Reviewer 3 Report

Dear authors, congratulations for your hard work. I have to admit that the issues that you are adresseing in your manuscript might be the least adressed, when it comes to this special category of people. We have all witnessed, more or less, the discriminations that members of the LGBTQ community are facing everyday, and having that said, I must admit that I find your research extremely interesting.

My major concern, in my humble opinion, regarding your methodology is the lack of control group. It is mandatory to have a control group, therefore I think that you should include a control sample, of men having sex with women and women having sex with men in order to have a baseline, otherwise the whole study lacks an necessary element.

I am under the impression that despite the limitations of your study that you mention, If you manage to provide data from a control group your study will surprise us in a positive way.

Author Response

Response to Reviewers:

We thank the reviewers for their thoughtful comments, which have helped us to strengthen the paper. Please see our point-by-point responses below. All page numbers referenced are for the tracked changes version of the revised manuscript. We have also made editorial changes throughout the paper to improve clarity.

  1. My major concern, in my humble opinion, regarding your methodology is the lack of control group. It is mandatory to have a control group, therefore I think that you should include a control sample, of men having sex with women and women having sex with men in order to have a baseline, otherwise the whole study lacks an necessary element. I am under the impression that despite the limitations of your study that you mention, If you manage to provide data from a control group your study will surprise us in a positive way.

Response:  We apologize for any confusion with the use of a reference group. We do have a reference group for descriptive findings, please see Table 1. In Table 1, we highlight some of the key differences in the outcomes (rates of SCD were slightly higher for TNB) and demographics. We then decided to limit final regression models to testing the associations between SCD and psychosocial factors among TNB participants only (n=115) given past research finding differences in the prevalence of SCD for TNB vs. cisgender LGBQ+, and our previous findings of higher rates of SCD in TNB adults compared TNB participants to LGBQ+ participants who did not identify as TNB. Please see page 4, lines 188-191

Round 2

Reviewer 3 Report

Dear authors please accept my congratulations for addressing all the issues that we mentioned before. Best of luck to all of you in your future work!